# LLMs as Scalable, General-Purpose Simulators For Evolving Digital Agent Training

## Abstract

Digital agents require diverse, large-scale UI trajectories to generalize across real-world tasks, yet collecting such data is prohibitively expensive in both human annotation, infra and engineering perspectives. To this end, we introduce **UI-Simulator**, a scalable paradigm that generates structured UI states and transitions to synthesize training trajectories at scale. Our paradigm integrates a digital world simulator for diverse UI states, a guided rollout process for coherent exploration, and a trajectory wrapper that produces high-quality and diverse trajectories for agent training. We further propose **UI-Simulator-Grow**, a targeted scaling strategy that enables more rapid and data-efficient scaling by prioritizing high-impact tasks and synthesizes informative trajectory variants. Experiments on WebArena and AndroidWorld show that UI-Simulator rivals or surpasses open-source agents trained on real UIs with significantly better robustness, despite using weaker teacher models. Moreover, UI-Simulator-Grow matches the performance of `Llama-3-70B-Instruct` using only `Llama-3-8B-Instruct` as the base model, highlighting the potential of targeted synthesis scaling paradigm to continuously and efficiently enhance the digital agents.

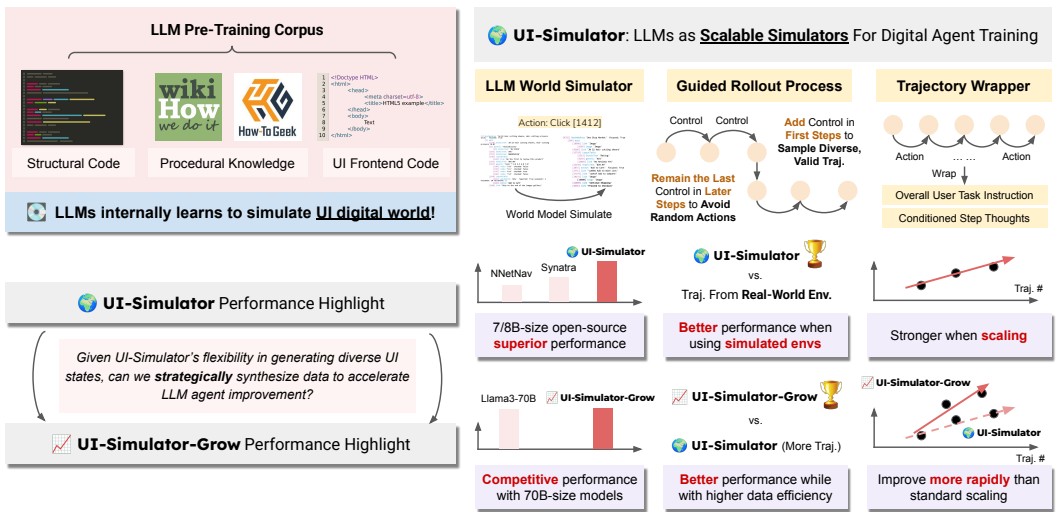

Figure 1: Overview and performance highlights of UI-Simulator and UI-Simulator-Grow.

## 1 Introduction

Large Language Models (LLMs) have emerged as the backbone of digital agents that follow user instructions and interact with diverse User Interface (UI) environments to accomplish complex tasks, such as daily web and mobile navigation (Deng et al., 2023; Koh et al., 2024; Zhou et al., 2024) and computer use tasks (Xie et al., 2024). A persistent bottleneck in training LLMs to become strong digital agents is the scarcity of large-scale, high-quality UI environment training trajectories. Collecting such data demands extensive human effort: for instance, Xie et al. (2024) report that

designing 360+ realistic computer-use tasks, which usually involve long, complex sequences of UI actions, requires more than 1,800 human hours. This cost severely limits the scalability of agent development and has sparked interest (Ou et al., 2024; Murty et al., 2024; Sun et al., 2024; Pahuja et al., 2025) in automatic synthesis of training trajectories.

When applying the automatic trajectory synthesis, what factors could significantly impact performance of the trained agent policies across different UIs? Motivated by Cobbe et al. (2020) and Kimi-K2 (Kimi et al., 2025), we argue that environment diversity would be a chief component, as exposing an agent to a wide variety of UI environments would increase its robustness and generalizability to unfamiliar tasks at test time. However, from infra and engineering aspects, deploying parallel real UI environments faces severe bottlenecks due to high resource demands, network instability, and the lack of native distributed support (Lai et al., 2025).

We notice that world models which model the environment states and their transitions (Munro, 1987; Ha & Schmidhuber, 2018) may offer a promising solution. If the world models enable the generation of diverse synthetic UI states, it will allow digital agents to immerse themselves in more diverse UI scenarios, enable richer rollouts and achieve stronger generalization to unseen apps and layouts. How can such digital world models be constructed? We argue that digital world models can be built on LLMs, as pre-training on front-end code and procedural knowledge makes them well-suited to synthesize realistic UI states and transitions triggered by user actions.

In this paper, we introduce **UI-SIMULATOR**, a scalable UI trajectory synthesis paradigm for training digital agents powered by an LLM-based digital world simulator. Given summaries of prior UI states and a next action, our **digital world simulator** generates future UI states in a hierarchical format without any additional fine-tuning. Each UI state encodes textual content, spatial coordinates, and dynamic attributes (e.g., focus status), organized into an accessibility tree structure that captures hierarchical relationships among regions and elements. To collect high-quality trajectories with UI-SIMULATOR, we run a step-wise guided rollout process where a teacher agent explores UIs generated by the world simulator under step-wise task control that prevents incoherent actions and promotes diverse, context-grounded behavior conditioned on prior actions and current state. Finally, a trajectory wrapper turns the rollouts into available training trajectories with user instructions, ground-truth UI actions, and step-wise reasoning.

Beyond simply blindly scaling up trajectory sizes with **UI-SIMULATOR** scaling, we explore how to *strategically* and *efficiently* synthesize data to accelerate LLM agent improvement. We introduce **UI-SIMULATOR-GROW**, a targeted scaling paradigm that achieves faster gains using fewer but more contributive trajectories. At each iteration, UI-SIMULATOR-GROW selects target tasks that offer the greater learning potential based on teacher-forcing loss signals from dynamically constructed validation sets, and synthesizes diverse trajectory variants to guide the next training iteration.

We evaluate UI-SIMULATOR on two widely used benchmarks, WebArena (Zhou et al., 2024) and AndroidWorld (Rawles et al., 2024), which cover web and mobile UI domains. UI-SIMULATOR achieves very competitive performance among open-source agents of comparable model size. Notably, UI-SIMULATOR is solely used to synthesize training resources with a weaker teacher model, `GPT-4o-mini`, whereas prior methods rely on the more powerful `GPT-4o` teacher model. We also find that UI-SIMULATOR yields greater robustness than other baselines when evaluated on perturbed UI environments, and higher adaptability given a limited amount of experience on test environment. Moreover, UI-SIMULATOR even outperforms the variants trained directly on real downstream environments using the same trajectory synthesis pipeline. Further, our targeted scaling paradigm UI-SIMULATOR-GROW using only `Llama-3-8B-Instruct` base model matches `Llama-3-70B-Instruct`, and drives steeper performance gains on WebArena using only 66% of the original training trajectories, demonstrating significantly improved data efficiency.

## 2 RELATED WORKS

**World Models.** Extensive prior work has explored learning dynamics models and using them to train decision-making policies (Werbos, 1987; Munro, 1987; Ha & Schmidhuber, 2018). Recently, the structural consistency of videos across tasks, environments, and embodiments has fueled progress in large-scale video pretraining for world models (Hafner et al., 2020; OpenAI, 2024; Parker-Holder et al., 2024). LLMs have also emerged as potential world models due to their rich encoding of

physical and textual knowledge from massive corpora. Hao et al. (2023); Gu et al. (2024); Chae et al. (2025) explore the use of LLMs as both reasoning agents and inference-time world models that proactively identify optimal actions. In our paper, we emphasize more scalable, high-quality UI trajectory synthesis and investigates the broader potential of digital world simulation for *agent training*. While prior work such as Fang et al. (2025); Gao et al. (2025) also utilizes LLMs as world models for agent training, our approach emphasizes greater efficiency of building digital simulators. Instead of training a dedicated world model, which can be costly due to the need for large-scale data, we directly leverage the LLM's prior knowledge, requiring little to no experience from downstream task environments. Moreover, our method supports a broader domain scope, extending beyond web interfaces to include mobile UIs. More importantly, we also introduce a targeted scaling paradigm that efficiently synthesizes the most contributive trajectories at each iterations, enabling faster agent improvement with significantly fewer resources.

**Synthetic Data for Digital Agent Training.** To overcome the bottleneck of limited high-quality human-annotated UI trajectories, recent efforts focus on scalable synthesis of training data for digital agents. Synatra (Ou et al., 2024) and AgentTrek (Xu et al., 2025) address this challenge by converting indirect human-readable knowledge (e.g., web tutorials and manuals) into direct task demonstrations, enabling large-scale supervision without manual annotation. NNetNav (Murty et al., 2024), OS-Genesis (Sun et al., 2024), InSTA (Trabucco et al., 2025), and Explorer (Pahuja et al., 2025) adopt unsupervised approaches that autonomously explore real-world websites and retroactively annotate action sequences with suitable instructions. Different from these methods, which rely solely on prior experience in real digital environments, UI-SIMULATOR leverages an LLM-based digital world model to simulate diverse, plausible, and previously unseen UI states which enable broader generalization and more robust agent training.

## 3 DIGITAL WORLD MODELS IN UI-SIMULATOR

In this section, we introduce how to build a digital world simulator fueled by LLMs in UI-SIMULATOR trajectory synthesis paradigm. The simulator construction process can be applied to a variety of digital and even non-digital agent tasks.

### 3.1 FORMULATION

We consider the task of simulating UI environment dynamics with LLMs to support agent training. LLMs can serve as the foundation of these simulators. Most digital UI environments, including web, mobile, and computer, can be represented as structured textual accessibility trees. Pre-training on front-end code and procedural knowledge makes LLMs suitable as a backbone model to synthesize reasonable UI states and state transitions triggered by user actions.

Let $s_t$ denote the full UI environment state at timestep $t$, $o_t$ be the corresponding observation visible to the agent, and $a_t$ be the corresponding action taken by the agent. Each element $e \subset s_t$ is associated with a bounding box $\text{bbox}(e) = (x_{\min}^e, x_{\max}^e, y_{\min}^e, y_{\max}^e)$, representing its position on the page. The environment dynamics are governed by a transition function $s_{t+1} = \mathcal{T}(s_t, a_t)$, where $\mathcal{T}$ is either the LLM used for digital world simulator $\mathcal{M}_{\text{LLM}}$ or rule-based transition function. The agent then receives a new observation $o_{t+1}$, computed by extracting the visible UI elements from $s_{t+1}$ based on the positions (see §A).

### 3.2 (RETRIEVAL-FREE) SIMULATION

To bridge the gap between our digital world simulation and real-world UI transition, we design a hybrid approach that considers rule-based and model-based transitions. Concretely, for most UI actions, our framework empowers the LLM $\mathcal{M}_{\text{LLM}}$ to generate realistic and diverse next-state transitions, serving as the core engine behind the simulator's ability to produce valid and imaginative UI states. The transition follows a multi-step pipeline that guides the world simulator to anticipate outcomes, infer coherent and diverse next states, and render them into a structured format.

**Predict an Overview of the Next State.** The first step in modeling the effect of an action is to generate a high-level overview of the next state, conditioned on the current state and the selected

Figure 2: Overall process of how the retrieval-free/-augmented simulators predict the next UI state.

action. For example, if the current state is a shopping website and the current action is typing the term *sneakers* into the search box and presses enter, the predicted overview of the next state would be, "*a search results page for the keyword sneakers*".

**Generate Rich Draft in Natural Language.** Based on the predicted overview, the LLM generates diverse and semantically rich content in natural language to populate the simulated webpage. The output of this step is intentionally unstructured and unconstrained by a fixed format, which encourages expressiveness and richness. The generated draft includes detailed descriptions of each element's attributes, such as the element's tag and content description, but without position information. For instance, a draft for a Reddit thread page might contain structural sections such as a heading section and a navigation section, as well as informative sections including the thread title, interaction area (e.g., upvote and comment buttons), and the main body of the post. This organization helps produce realistic and contextually rich simulated UI states in the next step.

**Convert Unstructured Draft into Structured Format.** We treat the LLM as a style transfer model that converts the unstructured natural language draft into a well-defined structured format, which can be directly used as training states in agent trajectories. During this process, coordinates are also automatically assigned to each UI element to complete the specification of $s_{t+1}$.

Note that some actions do not result in a completely new page but rather alter the view of the current state, e.g., a `scroll` action reveals content initially off-screen. To simulate deterministic actions with relatively fixed outcomes, we adopt rule-based transitions. See Appendix C for details.

### 3.3 Retrieval-Augmented Simulation

A common and realistic way to evaluate agent capability is by assessing how quickly it can adapt to a new test environment after limited experiences. Beyond the setting where no prior knowledge about the test environment is available, we also consider scenarios where **a small amount of the test environment experience is known**. For this scenario, we also introduce retrieval-augmented simulation, which conditions UI generation on limited experience from the test target environment. Compared to relying solely on the LLM world simulator's internal knowledge, this approach allows the world simulator to generate UI environments that not only resemble the target domains but also support a diverse range of tasks grounded in those environments. See Appendix D for more details.

## 4 Scalable Synthetic Training Trajectory Collection in Simulated World

In this section, we detail how LLMs are used to autonomously and continuously explore the digital world simulated by the LLM-based world model, generating high-quality training trajectories through guided rollouts and a final trajectory wrapper.

### 4.1 Overview and Formulation

We formulate our data collection process in two stages. The first stage is an *instruction-free* rollout process, where a teacher agent interacts with the LLM-based digital world simulator to generate synthetic trajectories without conditions on any predefined user instruction $G$. This goal-free setup

allows more flexible and executable trajectory synthesis unconstrained by specific task types. At each timestep $t$, given the current environment state $s_t$, observation $o_t$, and prior action history $H_t = [a_0, a_1, \ldots, a_{t-1}]$, the teacher agent $\mathcal{M}_{\text{Teacher}}$ samples the next action as $a_t \sim \mathcal{M}_{\text{Teacher}}(o_t, H_t)$. The environment then transitions to the next state $s_{t+1}$ via the world simulator LLM $\mathcal{M}_{\text{LLM}}$ prediction and deterministic static rules. The teacher continues the rollout until it determines that a semantically coherent task has been completed. In the end, we retrospectively derive a user task instruction $G$ that summarizes the intent underlying the completed trajectory. Each collected trajectory is then represented as $\tau = [o_0, a_0, o_1, a_1, \ldots, o_T, a_T]$, where $T$ is the trajectory length.

Scaling the collection methods across multiple environments and teacher rollouts yields a large, diverse training dataset for downstream UI agent policy learning. However, this procedure still leaves two critical questions: 1) How can we ensure both *diversity* and *validity* of the trajectories generated without explicit user instructions? 2) How can we generate *plausible and goal-consistent* user instructions $G$ that accurately reflect the completed trajectories? To this end, we propose a step-wise guided rollout process and a final trajectory wrapper to address these challenges.

## 4.2 Step-Wise Guided Rollout Process

We notice that without proper guidance for each rollout step, LLMs often exhibit biased behavior, leading to homogeneous tasks and trajectories. To mitigate the bias and increase the diversity and quality of our training set, we introduce a step-wise guided rollout process which proposes task controls to encourage exploration towards diverse yet reasonable directions. The pipeline involves the following steps (See more details in Appendix L):

**Step-Wise Task Control Proposal.** At first we prompt the teacher agents to envision common daily tasks users might perform based on the initial state, regarded as *first-step task control*. Specifically, given an initial state $s_0$, we prompt the $\mathcal{M}_{\text{Teacher}}$ to propose a high-level task description, referred to as the mentioned *task control*, $c_0 = \mathcal{M}_{\text{Teacher}}(s_0)$. For example, if $s_0$ is the home page of a shopping website, some examples of $c_0$ are "*Search for a certain product*" or "*Navigate to my account page*". When the actions related to first-step control were finished, the second-step control is updated based on the current observation, and this process continues iteratively. In general, suppose the trajectory just reaches its $t$-th step, under the $i$-th control. We define a boolean function $\text{Done}(c_i) \in \{\text{True}, \text{False}\}$ that indicates whether the current control $c_i$ has been completed, which is judged by $\mathcal{M}_{\text{Teacher}}$. The control update rule is given by:

$$c_i = \mathcal{M}_{\text{Teacher}}(s_t, \boxed{c_{i-1}}), \text{if } \text{Done}(c_{i-1}) = \text{True}.$$

For example, after the $\text{Done}$ function verified that the first-step control "*Navigate to my account page*" on the shopping website is completed, the proposed next-step control on the new *My Account* page in the shopping domain, could be generated as "*Check out my order history*", "*Modify my address information*", etc. This iterative goal proposal enables complex tasks to be compiled based on semantically meaningful sub-goals.

**Thought & Action Generation.** Under the current step's task control $c_i$ and prior rollout history $H_t$, the teacher agent $\mathcal{M}_{\text{Teacher}}$ is also prompted to produce its internal reasoning thought $r_t$, along with an action $a_t$ and the corresponding step summary $h_t$ in a CoT manner. Each thought provides a justification or plan for why the action is appropriate, and is recorded in the rollout history along with the step summary and action, $r_t, a_t, h_t \sim \mathcal{M}_{\text{Teacher}}(o_t, c_i, H_t)$, $H_{t+1} = H_t \cup [r_t; a_t; h_t]$, where ; indicates concatenation. To avoid endless rollouts, we allow $\mathcal{M}_{\text{Teacher}}$ to autonomously decide when to terminate the trajectory. That is, the agent will generate a $\text{STOP}$ action if it considers the task as completed, based on the current state and task control.

## 4.3 Trajectory Wrapper

The trajectory wrapper is designed to transform raw rollout trajectories into high-quality instances by inferring valid user instructions and reconstructing a coherent, step-by-step reasoning process. Since the rollout process is initially not guided by an explicit user instruction, in our trajectory wrapping process, we first use a task summarizer to condense the agent's actions into a concise description of what was accomplished and then further convert it as the final user instruction for the entire trajectory, denoting as $G$. To align the trajectory's reasoning with this synthesized instruction, we then ask the

teacher agent $\mathcal{M}_{\text{Teacher}}$ to rewrite and refine their thoughts, ensuring they are well-conditioned on the newly generated instruction and reflect the agent's internal decision-making.

Besides, reasoning ability is often a critical component in agent tasks, e.g., tasks like "*Tell me the most recent canceled order in the order history.*" To support this capability, we allow $\mathcal{M}_{\text{Teacher}}$ to insert intermediate reasoning thoughts when it deems such reasoning necessary or beneficial for conducting information query or analysis in the current UI state. In the end, we filter out low-quality trajectories based on the criteria: 1) actions must target valid elements and lead to meaningful state changes; and 2) the action mentioned in each step's reasoning should match the action actually taken.

## 5  UI-SIMULATOR-GROW: UI-SIMULATOR-POWERED TARGETED SCALING

In §4, we introduced UI-SIMULATOR for synthesizing diverse training trajectories. Rather than blindly increasing trajectory volume, we also explore how to *strategically* and *efficiently* scale to accelerate agent improvement. We propose **UI-SIMULATOR-GROW**, a **UI-SIMULATOR**–empowered targeted scaling paradigm that achieves faster gains with fewer synthesized trajectories.

UI-SIMULATOR-GROW iteratively identifies which tasks, if synthesized at the current stage, would most effectively enhance agent performance, and generates diverse trajectories for those tasks and their variants in preparation for the next training phase. In the first iteration, UI-SIMULATOR-GROW collects an initial batch of trajectories following the procedure in §4 as UI-SIMULATOR. In subsequent iterations, it automatically selects target tasks based on dynamically updated validation signals, synthesizes relevant trajectories, and applies continual learning to ensure steady performance gains. We introduce this in detail as follows.

**Target Task Selection.**  Target tasks for the next training iteration must satisfy the criteria: The tasks must be neither trivial for the current agent to solve, nor beyond the agent's present capabilities. Tasks the agent is already good at will offer limited learning signal, while tasks that are too hard may not lead to meaningful progress. We identify such target tasks by measuring the teacher-forcing loss of a teacher agent $\mathcal{M}_{\text{Teacher}}$ on the current validation set. Specifically, for each step, we treat the teacher's prediction as ground truth, compute the cross-entropy loss against the student agent's prediction, and average the loss over all steps for the tasks. Tasks are then ranked by loss, and those within the 25%–75% percentile range are selected as targets to filter excessively easy or hard tasks. See more cases in Appendix G.

As the agent improves after each iteration, the validation set is also updated to reflect the agent's evolving capabilities. In the first iteration, it is an independent batch of tasks synthesized in the same way as the initial training set. For later iterations, we consider two construction strategies – *Strategy 1: The validation set is composed entirely of a split from the newly synthesized data for the upcoming iteration*; *Strategy 2: The validation set includes 50% of trajectory variants from the previous iteration's target tasks and 50% from new tasks*. Both strategies aim to promote continual improvement and prevent future iteration evaluation from overfitting to earlier iterations.

**Synthesizing Diverse Target Task Trajectory Variants.**  After identifying target tasks that can effectively challenge the agent, we synthesize additional trajectories focused on those tasks. One strategy we adopted is lightweight task rewriting, where the task instruction is slightly modified without changing its core structure or logic. The corresponding environment states, thoughts, and actions are adjusted accordingly, while preserving the overall reasoning flow. For example, a selected task like "*search running shoes*" might be rewritten as "*search slippers*". Since the task logic remains consistent, the UI states and actions (e.g., entering a query, clicking a result) are similarly structured. We prompt the LLM to maintain the task's action types and flow, modifying only content-specific elements in the UI states such as product names. This ensures meaningful variation while preserving alignment with the agent's learning objectives.

**Continual Learning.**  As UI-SIMULATOR-GROW continuously incorporates new synthesized training trajectories through iterations, a key challenge is adapting the agent policies without forgetting. We address this with continual learning (Biesialska et al., 2020), focusing on replay methods, a widely used technique that revisits selected tasks from prior training stages.

Table 1: Overall success rate (SR) on the WebArena and AndroidWorld test sets. $<<$ indicates methods with substantially less exposure to the real downstream test environments.

| Models | Teacher Agents | Train Under Real Env.? | WebArena SR (%) | AndroidWorld SR (%) |
|---|---|---|---|---|
| Base Open-Source LLMs and Proprietary LLMs | | | | |
| Llama-3-8B-Instruct | - | ✗ | 2.34 | - |
| CodeLlama-34B-Instruct | - | ✗ | 4.06 | - |
| Lemur-chat-70B | - | ✗ | 5.30 | - |
| Llama-3-70B-Instruct | - | ✗ | 7.02 | - |
| Gemini Pro | - | ✗ | 7.12 | - |
| Qwen-1.5-72B-Instruct | - | ✗ | 7.14 | - |
| Qwen-2.5-7B-Instruct | - | ✗ | 3.94 | 0.0 |
| Qwen-2-VL-7B | - | ✗ | - | 5.0 |
| Qwen-2-VL-72B | - | ✗ | - | 5.0 |
| Gemma-2-27B | - | ✗ | - | 9.5 |
| GPT-4o | - | ✗ | 13.10 | 11.7 |
| Digital Agent Training Data Synthesis Baselines | | | | |
| AgentFlan | N/A | ✓ | 4.68 | - |
| NNetNav | Llama-3.1-70B | ✓ | 4.80 | - |
| Synatra | GPT-4-turbo | ✓ | 6.28 | - |
| OS-Genesis | GPT-4o | ✓ | 6.16 | 9.1 |
| GUIMid (Post-Train) | N/A | ✓ | 6.20 | 9.0 |
| UI-SIMULATOR-Series Variants | | | | |
| UI-SIMULATOR-F | GPT-4o-mini | ✗ | 6.28 | 8.6 |
| UI-SIMULATOR-R | GPT-4o-mini | ✓ ($<<$) | **6.40** | **12.9** |
| UI-SIMULATOR-GROW-R | GPT-4o-mini | ✓ ($<<$) | **7.14** | **13.4** |

Following Dynosaur (Yin et al., 2023), we adopt a replay strategy that selects the most representative tasks from the previous iteration. Given $N$ tasks from the prior iteration, we use Sentence Transformer (Reimers & Gurevych, 2019) based on RoBERTa-large (Liu et al., 2019) to encode task instructions into a matrix $I_p \in \mathbb{R}^{N \times d}$, where $d$ is the embedding dimension. We then compute cosine similarities: $S_{pp} = \cos\_\mathrm{sim}(I_p, I_p) \in \mathbb{R}^{N \times N}$. Tasks with the top row sums in $S_{pp}$ are representative and selected for replay.

# 6 EXPERIMENTS

## 6.1 EXPERIMENTAL SETUP

**Evaluation Benchmarks.** We evaluate the LLM agents trained with UI-SIMULATOR on two benchmarks across web and mobile domains: **WebArena**, which contains 812 complex yet realistic web navigation tasks; and **AndroidWorld**, which consists of 116 challenging daily mobile usage tasks. We report the **success rate** (SR) across all tasks. The temperature for model inference is set to 0.6; preliminary experiments suggest that varying the temperature does not significantly affect performance. Note that for a fair comparison, we reproduce and evaluate those methods under the original WebArena evaluation settings[1], instead of BrowserGym or any lite versions.

**Digital World Simulation, Trajectory Collection and Agent Training.** We use `GPT-4o-mini` for both state simulation and guided rollout. For WebArena, we train our digital agents and baseline agents using `Llama-3-8B-Instruct` as the base model. For AndroidWorld, we choose to use `Qwen-2.5-7B-Instruct` due to its extended context length support beyond 8192 tokens, a common requirement in AndroidWorld which exceeds the maximum context length of `Llama-3-8B-Instruct`. More details are discussed in Appendix E and I.

## 6.2 OVERALL PERFORMANCE

Results are presented in Table 1. We denote the UI-SIMULATOR variants without and with retrieval-augmented simulation as UI-SIMULATOR-F and UI-SIMULATOR-R, respectively.

**UI-SIMULATOR-F.** We observe that even without exposure to real-world test environments, training solely on LLM-simulated environments can significantly enhance its base model's performance. This is particularly evident on AndroidWorld, where the success rate increases from 0% to 9%. UI-SIMULATOR-F even outperforms OS-Genesis on WebArena, which is trained using trajectories

---

[1]The same as https://github.com/web-arena-x/webarena.

synthesized directly from the WebArena test environments. These show that LLMs possess sufficient knowledge to generate reliable and coherent digital environment simulations, offering a promising alternative when real test environments involve high latency or are difficult to access.

**UI-SIMULATOR-R vs. Larger & Proprietary Models.**  We observe that UI-SIMULATOR-R performs on par with `Gemini-Pro` on WebArena, as well as `GPT-4o` on AndroidWorld, despite being built on a much smaller 8B-scale LLM. This highlights the strong generalization capability of UI-SIMULATOR, even with limited exposure to the target environment.

**UI-SIMULATOR vs. Open-Source Agent Traj. Synthesis Baselines.**  UI-SIMULATOR-R surpasses OS-Genesis, which relies on the stronger `GPT-4o` teacher to generate training trajectories within test environments, while even UI-SIMULATOR-F achieves superior performance on Android-World despite being trained only with trajectories from the weaker `GPT-4o-mini`. These results highlight the potential of UI-SIMULATOR when paired with stronger teacher agents. Moreover, unlike NNetNav and OS-Genesis, which generate synthetic training data through extensive unsupervised interaction with the test environments, UI-SIMULATOR-R restricts the environment exposure to a much smaller scope. Despite this, it still outperforms NNetNav and OS-Genesis by 2.2% and 0.9% on WebArena, and surpasses OS-Genesis by 3.8% on AndroidWorld, demonstrating the effectiveness of our simulation-driven approach in enabling rapid adaptation to test environments.

## 7 ANALYSIS

In this section, focusing on WebArena, we conduct a comprehensive analysis to evaluate the advantages and potential of the UI-SIMULATOR framework. We present a series of training experiments alongside qualitative studies to examine each core component of UI-SIMULATOR and illustrate how UI-SIMULATOR-GROW helps effective scaling. More human evaluation on synthesized training trajectory quality is discussed in Appendix F and J.

### 7.1 ABLATION STUDY

**Agent Robustness Brought From UI-SIMULATOR.**  Thanks to the flexibility of UI-SIMULATOR in generating diverse UI layouts, agents trained on its synthesized trajectories gain robustness to varied UI states. To test this, we perturb WebArena and AndroidWorld UI states by randomly shuffling layout structures while preserving UI content, ensuring validity and identical solutions. For a meaningful comparison with other strong baselines, we focus on comparison with the baseline OS-Genesis, whose performance is closest to ours on both datasets. We notice that UI-SIMULATOR-F, which synthesizes UI states without referencing downstream environments, suffers the smallest performance change.

**Simulated Digital Environment vs. Real Test Environment.**  What happens if we collect similar number of training trajectories directly on the real test environments? Surprisingly, UI-SIMULATOR can even outperform this strong baseline. We identify one key reason for this performance gap: the real test environments may not be able to consistently provide useful state transitions or cover diverse interaction scenarios. For example, if the search query keywords do not match any entries in a shopping website, the environment may return *Search not found*, and such search tasks are likely to be excluded; For tasks involving account pages, if the user is not logged in beforehand, those trajectories cannot be collected due to access restrictions. In contrast, both tasks can be easily synthesized in the digital world model without any such constraints. This highlights the potential of UI-SIMULATOR to go beyond the limitations of real environments by generating trajectories that are infeasible to obtain.

**UI-SIMULATOR-R vs. OS-Genesis Sharing the Same Amount of Prior Experience on Test Environment.**  Both UI-SIMULATOR-R and OS-Genesis have access to the test environment; however, OS-Genesis benefits from significantly more experience on the test environments. To assess them under equal conditions, we control for the amount of test environment experience and compare their performance. On both WebArena and AndroidWorld, we even find that UI-SIMULATOR-R achieves around 4 and 2.5 times the performance of OS-Genesis, respectively, highlighting its ability to generate highly adaptive digital agents even with limited exposure to the real environment.

**Rollout and Simulation Process Design.** We ablate our rollout and simulation design, both key to synthesizing high-quality trajectories, and report details in Appendix Table 6 (see Appendix F). Removing step-wise task controls lowers performance by 4.7% and significantly reduces trajectory diversity. Besides, replacing multi-step with single-step simulation incurs a further 2.4% drop. Multi-step simulation encourages the world model to *think step-by-step* and output rich content by first sketching out what the effect of the current action and the potential draft for the next state, resulting in trajectories with higher quality. These results show that fine-grained, step-wise control and multi-step simulation are both essential for generating high-quality trajectories.

| Models | WA (%) | AW (%) |
|---|---|---|
| UI-SIMULATOR-F | **6.28** | 8.6 |
|     Perturbed Env. | 5.54 | **8.7** |
|     Synthesize in Real Env. | 4.31 | 4.7 |
| UI-SIMULATOR-R | **6.40** | **12.9** |
|     Synthesize in Real Env. | 4.31 | 4.7 |
| OS-Genesis | 6.16 | 9.1 |
|     Perturbed Env. | 4.43 | 8.7 |
|     Same # of Experience | 1.48 | 5.2 |

Table 2: Ablation study on robustness, synthesize trajectories from real test environment and utilization of the same amount of experience. WA and AW are WebArena and AndroidWorld in short.

## 7.2 UI-SIMULATOR-GROW VS. STANDARD UI-SIMULATOR SCALING

We compare the UI-SIMULATOR-GROW with the standard UI-SIMULATOR scaling to examine which paradigm more effectively accelerates agent performance. For standard scaling, we use the full UI-SIMULATOR-R training set split it into three equal parts, and emulate a 3-iteration scaling process by progressively adding one more split at each iteration. For UI-SIMULATOR-GROW, the first iteration uses the same initial split as in standard scaling. Subsequent iterations of UI-SIMULATOR-GROW rely primarily on synthesized variants of target tasks

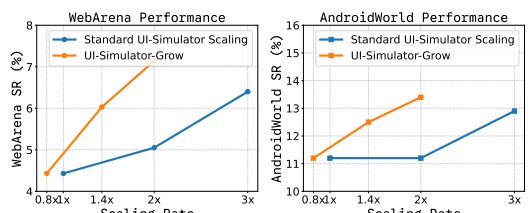

Figure 3: The effect of standard scaling and UI-SIMULATOR-GROW targeted scaling.

selected from the last iteration, supplemented by portions of the remaining splits for constructing dynamic validation sets, instead of blindly adding more new generic trajectories as standard scaling does. Note that the process guarantees that UI-SIMULATOR-GROW draws only from UI-SIMULATOR-R and its generated variants, without introducing any external data beyond its scope for fair comparison.

As shown in Figure 3, UI-SIMULATOR-GROW yields a steeper performance improvement than standard scaling. Notably, by the third iteration, it matches the performance of Qwen-1.5-72B-Instruct and surpasses Llama-3-70B-Instruct. Moreover, UI-SIMULATOR-GROW uses only 66% of the original UI-SIMULATOR-R trajectories, demonstrating more efficient data utilization compared to indiscriminate trajectory generation. Beyond overall success rates, we examine how performance improves under UI-SIMULATOR-GROW. In Appendix H, we further show that UI-SIMULATOR-GROW consistently avoids performance degradation across web task categories, and in some cases enables success on task types where standard scaling always fails. We attribute these gains to the combination of dynamic validation set construction and targeted task variant synthesis, which guide the agent toward underperforming task types and reinforce its capabilities in a focused manner.

## 8 CONCLUSIONS

We introduced UI-SIMULATOR, a scalable trajectory synthesis paradigm that uses LLM-based digital world simulators to synthesize diverse UI trajectories at scale through multi-step simulation, guided rollouts, and final trajectory wrapping. We further propose UI-SIMULATOR-GROW, a targeted scaling paradigm that prioritizes high-impact tasks for more data-efficient continuous improvement. Experiments on WebArena and AndroidWorld show that UI-SIMULATOR rivals or surpasses real-environment training despite using weaker teacher agents, while UI-SIMULATOR-GROW achieves more rapid improvement trend than standard UI-SIMULATOR scaling with only 66% training data and even matches 70B-scale models. Ablation study further highlight the promise of simulation-driven trajectory synthesis as a more adaptive and robust approach for advancing digital agents. Beyond extending to other UI domains such as desktop, our future work envisions applying the world simulator and targeted scaling method to any environment representable in text.

## REPRODUCIBILITY AND LLM USAGE STATEMENT

We will release the source code and training data publicly upon publication. LLMs were only used for editorial polishing of the manuscript and did not contribute to research ideation.

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

APPENDIX

## A    OBTAINING OBSERVATION $o_t$ FROM UI STATE $s_t$

As defined in §3.1, the observation $o_t$ is obtained by extracting the UI elements visible within the current state $s_t$. Let the viewport at timestep $t$ be

$$\mathcal{V}_t = [x_0, x_1] \times [y_0, y_1].$$

The observation at step $t$ is then calculated by,

$$o_t = \{e \in s_t \mid \text{bbox}(e) \cap \mathcal{V}_t \neq \emptyset\},$$

i.e., capturing the set of elements whose bounding boxes intersect with the viewport region.

## B    ACTION SPACES IN UI SIMULATOR

As shown in Table 3 and 4, we summarize the supported action spaces for WebArena and Android-World, which span the common UI interactions (e.g., click, type, scroll).

Table 3: Action space of WebArena environment.

| Action | Description |
|---|---|
| `click [id]` | Click the element with the given ID. |
| `type [id] [content] [0|1]` | Type `content` into the text box with the specified ID; press Enter if the last argument is `1`. |
| `hover [id]` | Hover over the element with the given ID (often to trigger popups or dropdowns). |
| `press [key_comb]` | Press a keyboard combination (e.g., `ctrl+c`). |
| `scroll [direction]` | Scroll the page in the specified direction: `up`, `down`, `left`, or `right`. |
| `go_forward` | Go forward to the next page (only after a prior `go_back`). |
| `go_back` | Go back to the previous page. |
| `new_tab` | Open a new empty browser tab. |
| `tab_focus [index]` | Switch focus to the tab at the given index. |
| `close_tab` | Close the current tab. |
| `goto [url]` | Navigate to the specified URL. |

Table 4: Action space of AndroidWorld environment.

| Action | Description |
|---|---|
| `click [id]` | Tap the element with the given ID on the current screen. |
| `open_app [app_name]` | Launch the app with the specified name. |
| `input_text [id] [content]` | Enter `content` into the text box identified by `id` using the keyboard. |
| `keyborad_enter` | Press the Enter key. |
| `scroll [direction]` | Scroll the screen in the specified direction: `up`, `down`, `left`, or `right`. |
| `navigate_back` | Go back to the previous screen. |
| `navigate_home` | Return to the app's home screen. |
| `wait` | Remain idle for a short duration. |

## C    DETAILS OF RULE-BASED TRANSITION

As discussed in §3.2, certain actions in the action space (e.g., `Type`, `Scroll`) involve deterministic state transitions. To better simulate this, we incorporate rule-based transitions that enhance realism in the simulation. The details of these rule-based transitions are introduced below.

**Type Action.** This is the most straightforward case. The simulator updates the target element by appending the typed content into its `content_description` attribute.

**Scroll Action.** The simulator will keep on the same state after taking a `Scroll` action, i.e., $s_{t+1} = s_t$. The simulator maintains a scroll offset $(x_{\text{offset}}, y_{\text{offset}}) \in \mathbb{R}^2$, which determines the visible region of the UI along with the window dimensions $(w_{\text{win}}, h_{\text{win}})$. Here we take $(w_{\text{win}}, h_{\text{win}}) = (2400, 1080)$.

Initially, the scroll offset is set to $(x_{\text{offset}}, y_{\text{offset}}) = (0, 0)$. When a `Scroll` action is performed, the scroll offset is updated as follows:

$$(x_{\text{offset}}, y_{\text{offset}}) \leftarrow (x_{\text{offset}} + \Delta x, \ y_{\text{offset}} + \Delta y),$$

where $(\Delta x, \Delta y)$ are the fixed scroll displacements in horizontal and vertical directions, respectively. For instance, `scroll [down]` corresponds to $(\Delta x, \Delta y) = (0, 1080)$

The observation at timestep $t$, denoted $o_t$, consists of all UI elements whose bounding boxes intersect with the current visible viewport:

$$\mathcal{V}_t = [x_{\text{offset}}, x_{\text{offset}} + w_{\text{win}}] \times [y_{\text{offset}}, y_{\text{offset}} + h_{\text{win}}],$$
$$o_t = \{e \in s_t \mid \text{bbox}(e) \cap \mathcal{V}_t \neq \emptyset\}.$$

After the scroll offset is updated, the observation $o_{t+1}$ is recomputed based on the new viewport $\mathcal{V}_{t+1}$, and the state itself remains unchanged.

**New_tab, Navigate_back, Navigate_forward Actions.** We model web browsing as a traverse process on the tree structure. To support this, the simulator maintains an explicit browsing stack to track session history. These actions deterministically alter the current tab state based on prior states stored in the stack.

## D  RETRIEVAL-AUGMENTED SIMULATION

As discussed in §3.3, we also study how quickly the simulator can adapt to a new test environment given only limited prior experience. *Retrieval-augmented simulation* addresses this by conditioning UI generation on a small set of real examples from the target environment, producing future states that resemble the target domain, coherent with the previous rollout steps, and support a diverse range of grounded tasks.

Formally, we first construct an *offline* retrieval corpus of $N$ state transitions from the test environment, denoted as, $\mathcal{D} = \left\{ \left( \tilde{o}_t^{(i)}, H_t^{(i)}, \tilde{o}_{t+1}^{(i)}, \tilde{s}_t^{(i)}, \tilde{s}_{t+1}^{(i)} \right) \right\}_{i=1}^{N}$, where $\tilde{o}_t^{(i)}$ and $\tilde{o}_{t+1}^{(i)}$ are the observations before and after an action in the downstream test environment; $\tilde{s}_t^{(i)}$ and $\tilde{s}_{t+1}^{(i)}$ are their corresponding UI states; and $H_t^{(i)}$ denotes the action history up to timestep $t$.

During UI-SIMULATOR paradigm, when simulating the next UI state after a given action, we query this *offline* retrieval corpus with the current observation–action history pair $(o_t, H_t)$. A retriever then returns the most relevant observation $\tilde{o}_{\text{ret}}$ and corresponding state $\tilde{s}_{\text{ret}}$ from the corpus $\mathcal{D}$. The transition can be modelled as $s_{t+1} = \mathcal{M}_{\text{LLM}}(s_t, a_t, \boxed{s_{\text{ret}}})$, where $\mathcal{M}_{\text{LLM}}$ is prompted with both the current interaction context and the retrieved state $s_{\text{ret}}$, grounding the simulation in prior experience while still allowing the creation of novel, coherent UI states. The key distinction from retrieval-free simulation is the incorporation of the retrieved state $s_{\text{ret}}$ into the simulation process.

In practice, we employ a hybrid retrieval pipeline over $\mathcal{D}$ to retrieve the transition that is most semantically similar to the current trajectory simulation. The retrieval process proceeds in three stages: **First**, a coarse filtering step is performed using BM25 ranking, where the *action history serves as the query*, to retrieve the transitions with very similar action histories. **Next**, we use *current action histories as the query* again to further narrow down the more relevant transitions from the transitions stored in $\mathcal{D}$ by utilizing `GPT-4o` as the semantic retriever, which captures deeper semantic

similarities with the current action history queries. **Finally**, we construct a composite retrieval key that incorporates both the current *state* and *action history*, and apply BM25 again to select the most relevant transition. Despite the small size of $\mathcal{D}$, this hybrid strategy still improves the consistency and realism of the generated UI states.

# E  KEY STATISTICS AND HYPERPARAMETERS OF STEP-WISE ROLLOUT PROCESS

Table 5: Step numbers of the collected trajectories and step-wise task control numbers across domains.

|  | Shopping | Gitlab | Map | Reddit | Shopping Admin | Android |
|---|---|---|---|---|---|---|
| Step # | 800 | 1500 | 1500 | 1300 | 1500 | 6500 |
| Step-wise task control # per proposal | 5 | 8 | 3 | 6 | 8 | 5 |

As we mentioned in §4, we introduce a step-wise guided rollout process to encourage exploration towards diverse yet reasonable directions for UI trajectory synthesis. Table 5 summarizes key statistics and hyperparameters of the rollout process. The first row reports the number of final synthesized trajectory steps for each domain, while the second row shows the number of task controls for rollout guidance when the teacher agent is going to propose such task controls. Variation in task control numbers reflects the complexity of website content: for instance, map websites are relatively simple, supporting only a few core functions such as search or navigation, whereas domains like Gitlab, Reddit, and shopping admin pages contain many more elements and support various functionalities, requiring more extensive task control.

In the web environment, we collect 2K trajectories with an average length of 3.3 steps. In the Android mobile environment, we collect 1.3K trajectories averaging 5 steps each. The collected states are adjusted to fit the domain and format as defined in WebArena and AndroidWorld benchmarks. For retrieval-augmented simulation, the collection process leverages a limited amount of experiences from the two environments. For the size of the offline retrieval corpus $\mathcal{D}$, we have 1647 transition experience for WebArena, and 683 for AndroidWorld (approximately only 25% and 10% of OS-Genesis experiences on test environments).

The estimated cost per web trajectory is \$0.02 for retrieval-free simulation and \$0.05 for retrieval-augmented simulation, and the estimated cost doubles for each AndroidWorld training trajectory.

# F  ADDITIONAL ABLATION STUDY ON TRAJECTORY COLLECTION PROCESS

Table 6: Ablation study performance about rollout process design on WebArena (WA) and Android-World (AW) with success rate as our metric.

| Models | WA (%) | AW (%) |
|---|---|---|
| UI-SIMULATOR-R | **6.40** | **12.9** |
| w/o Step-Wise Task Control | 1.72 | 5.2 |
| w/o Multi-Step Simulation | 4.06 | 9.1 |

In this section, we discuss more ablation study on the step-wise guided rollout process design to justify the effectiveness of our design.

**Step-Wise Task Control.**   In this experiment, we remove all step-wise task controls to collect a new set of training trajectories, aiming to assess whether the guided rollout process contributes to generating higher-quality data. From Table 6, we observe a performance drop of around 4.7% and 7.7% on WebArena and AndroidWorld following the removal of task controls. Upon closer inspection of the newly collected trajectories, we find that trajectory diversity suffers significantly. Without user instructions as conditioning signals, the teacher agent tends to repeatedly sample the same one or two elements due to inherent model biases. This further highlights the importance and effectiveness of fine-grained, step-wise control in our trajectory collection process.

**Multi-Step Simulation.** In this experiment, we replace multi-step simulation with single-step simulation to examine whether the simplified approach can still yield satisfactory simulations and benefit downstream tasks. As shown in Table 6, this modification results in a performance drop of approximately 2.4% and 3.8% on WebArena and AndroidWorld. Single-step simulation, though cost-saving, would always generate common, biased content, thereby harming the diversity and content richness of the collected training set. In contrast, we encourage the world model to output rich, diverse content by splitting the simulation into multiple steps, resulting in trajectories with higher quality.

## G ANALYSIS ON TARGETED TASK SELECTION IN UI-SIMULATOR-GROW PARADIGM

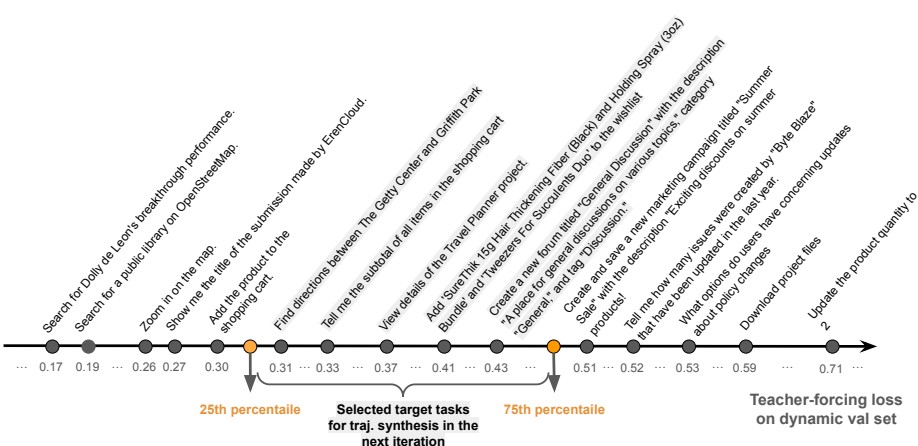

Figure 4: Illustration of overall target task selection process.

In this section, we describe how target tasks are selected for synthesizing new training trajectories in the next iteration of the UI-SIMULATOR-GROW paradigm. Figure 4 illustrates the task selection process between the first and second UI-SIMULATOR-GROW iterations for WebArena. We begin by ranking all tasks in the current validation set by their teacher-forcing loss, from smallest to largest. Tasks below the 25th percentile are considered too easy or already well learned (e.g., zooming on a map or searching for a location) and are excluded from further synthesis. Conversely, tasks above the 75th percentile are often overly challenging or ambiguous, and are likewise excluded. The remaining middle range of tasks is chosen as the target set for the next training iteration.

## H QUALITATIVE ANALYSIS ON SUCCESSFUL TASKS THROUGH UI-SIMULATOR-GROW SCALING

Beyond overall success rates, we closely examine how performance improves under UI-SIMULATOR-GROW. As shown in Figure 5, a consistent upward trend appears across most major WebArena task categories. Notably, for code repository operations, the final iterations of UI-SIMULATOR-GROW demonstrate the ability to solve tasks that neither standard UI-SIMULATOR scaling nor earlier iterations could handle. This highlights the paradigm's potential to enable agents to tackle increasingly diverse and complex tasks.

## I TRAINING AND EVALUATION DETAILS

For digital UI state simulation, the LLM-based world simulator are run with a decoding temperature of 0.5. During the trajectory synthesis process, teacher agents also generate next action with the decoding temperature of 0.5.

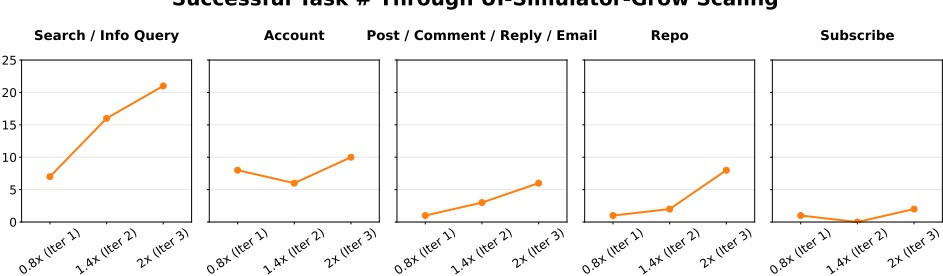

Figure 5: Successful task numbers across the 5 main task categories through the three iterations of the UI-SIMULATOR-GROW scaling.

Table 7: Human evaluation dimensions with definitions and illustrative examples.

| Dimensions | Definitions | Examples |
|---|---|---|
| **Realism of Task** | Whether the task resembles something a real user would encounter in everyday app usage. | "Search for a product and add it to the cart" is realistic; "click random buttons" is not. |
| **State Reasonability** | Whether the UI states and their transitions are reasonable given the app's typical structure and context. | A "checkout" button inside a map application is unreasonable. |
| **Action Validity** | Whether each action logically corresponds to the goal, the current state, and the intended next state. | Clicking "submit" should occur only after all required entries are filled. |
| **Logical Consistency (Thoughts)** | Whether explanatory comments or inferred logic are coherent and free of contradictions. | "User clicks search to find item" followed by "user wants to delete profile" is inconsistent. |
| **Task Completion** | Whether the trajectory ends with the task's goal fully achieved. | If the goal is "send a message," the message should be sent in the final step. |
| **Trajectory Consistency** | Whether actions and transitions form a coherent flow, with no contradictions or unexpected diversions. | The trajectory should not jump between unrelated tasks or contexts. |
| **# Irrelevant Steps** | Number of steps unrelated to the goal; a high count indicates inefficiency or redundancy. | Clicking "About Us" is irrelevant to "creating an account." |
| **Topic Abstraction** | Whether the task is generalized and meaningful, not just low-level UI manipulation. | "Complete login" is abstracted; "click input, type name, click button" is not. |

We train `Llama-3-8B-Instruct` and `Qwen-2.5-7B-Instruct` for WebArena and Android-World, respectively, using a batch size of 48, learning rate $1 \times 10^{-5}$, and 2 epochs. Training is performed on 4 A6000 GPUs (48GB each) with Liger-Kernel (Hsu et al., 2025) to improve throughput and reduce memory usage.

During inference on the downstream benchmarks, we set the generation temperature to 0.6 and the maximum output length to 1024 tokens.

## J    HUMAN EVALUATION OF TRAINING TRAJECTORIES SYNTHESIZED BY UI-SIMULATOR

Beyond quantitative metrics which already demonstrate the effectiveness of training trajectories synthesized by UI-SIMULATOR, we further conduct a qualitative human evaluation of the training trajectories across 8 dimensions (Table 7). For each dimension, scores are computed as the proportion of trajectories that satisfy the corresponding evaluation criterion.

We recruited three annotators, each holding a master's degree or higher in computer science. Each annotator evaluated 40 trajectories from both UI-SIMULATOR-F and UI-SIMULATOR-R. We built a

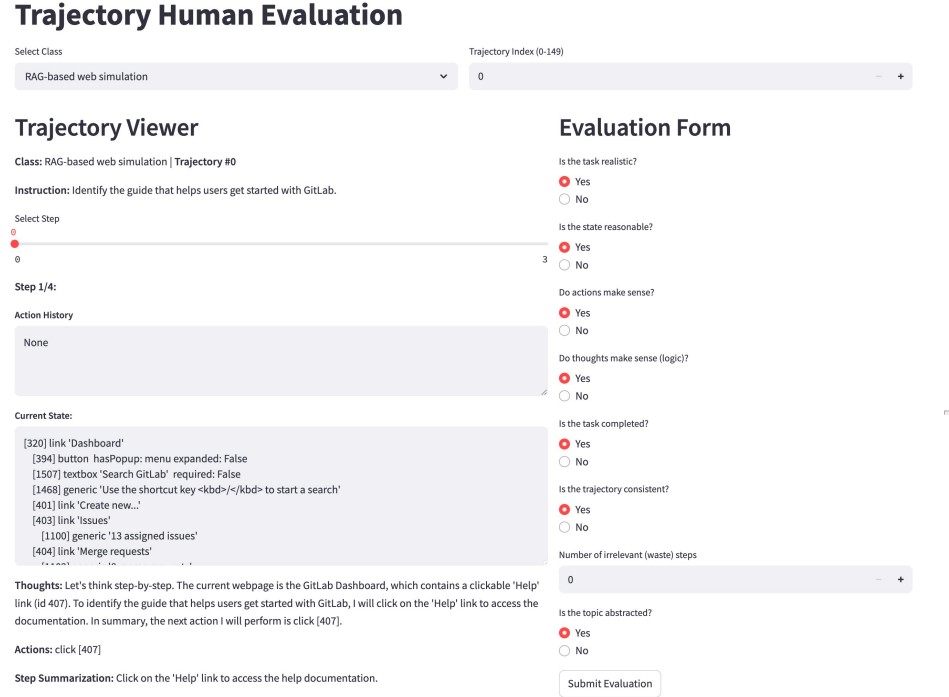

Figure 6: The front-end web interface for trajectory human evaluation.

Table 8: Average human evaluation scores across dimensions (Web).

| Dimensions | UI-SIMULATOR-R | UI-SIMULATOR-F |
|---|---|---|
| **Realism of Task** | 0.914 | 0.942 |
| **State Reasonability** | 0.952 | 0.875 |
| **Action Validity** | 0.867 | 0.767 |
| **Logical Consistency (Thoughts)** | 0.867 | 0.733 |
| **Task Completion** | 0.938 | 0.908 |
| **Trajectory Consistency** | 0.971 | 0.917 |
| **# Irrelevant Steps** | 0.214 | 0.533 |
| **Topic Abstraction** | 0.990 | 1.000 |

front-end website for annotating, as shown in Figure 6. To assess reliability, we measured agreement on 30 overlapping trajectories, yielding pairwise scores of 0.876, 0.890, and 0.976, which indicate strong consistency.

Table 8 and 9 present the average human evaluation scores across all dimensions. We observe that satisfaction rates for each dimension consistently reach, and in many cases exceed 90%. It suggests that even without additional fine-tuning, LLMs are already capable of serving as effective digital world simulators for scaling high-quality training trajectory synthesis.

## K   UI SIMULATION ISSUE ANALYSIS

While the digital world simulator significantly enhances agent training, it may still exhibit minor discrepancies in capturing certain real-world UI state transitions. Figure 7 and 8 demonstrate two cases where UI-SIMULATOR-F and UI-SIMULATOR-R may not simulate well.

Figure 7 shows a transition where UI-SIMULATOR-F mistakenly fuses irrelevant context into the next step simulation. The new webpage should be a list of all available forums in realistic Reddit after clicking the `Forums` link. However, UI-SIMULATOR-F makes an error by taking the context of

Table 9: Average human evaluation scores across dimensions (Android).

| Dimensions | UI-SIMULATOR-R | UI-SIMULATOR-F |
|---|---|---|
| **Realism of Task** | 0.884 | 0.888 |
| **State Reasonability** | 0.873 | 0.888 |
| **Action Validity** | 0.856 | 0.806 |
| **Logical Consistency (Thoughts)** | 0.884 | 0.847 |
| **Task Completion** | 0.862 | 0.929 |
| **Trajectory Consistency** | 0.939 | 0.959 |
| **# Irrelevant Steps** | 1.09 | 0.794 |
| **Topic Abstraction** | 1.0 | 0.994 |

current forum, `\f\deeplearning`, into account. As a result, the new webpage shows a bunch of information related to deep learning.

Transition in Figure 8 shows that UI-SIMULATOR-R sometimes ignores the current context and refers too much to the retrieved reference state. The search results of `Byte Blaze` should be relevant to the keyword. However, in this case UI-SIMULATOR-R just simulates the search results from the reference state and ignores what the user is currently searching.

Figure 7: A case of failed simulation where UI-SIMULATOR-F generates the new page based on irrelevant context.

## L    SYSTEM PROMPTS

In this section, we present the key system prompts used in our work.

Tables 10–16 provide the prompts for the guided rollout process, Tables 17–20 detail those used during simulation, and Table 21 illustrates how we generate targeted tasks for UI-SIMULATOR-GROW.

The prompts used for simulating Android and web UI environment states are largely aligned. Table 22 presents the system prompt for thought and action generation in Android trajectory collection, whose instructions closely mirror those used for the web setting. All states are formatted according to the AndroidWorld specification, represented as lists of UI elements with attributes such as `content_description` and `class_name`.

```
[320] link 'Dashboard'                          [6320] RootWebArea 'Search results for "Byte
    [394] button  hasPopup: menu expanded:            Blaze" · GitLab'  focused: True
        False                                   [9128] link 'Projects 7'
    [1507] textbox 'Search GitLab'  required:   [5946] link 'Milestones 4'
        False                                   [15074] link 'Users 15'
    [1468] generic 'Use the shortcut key        [2986] link 'O'
        / to start a search'         [4708] heading 'Open Source Diversity /
    [401] link 'Create new...'                       opensourcediversity.org'
    [403] link 'Issues'                             [12492] link 'Open Source Diversity /
        [1100] generic '13 assigned issues'              opensourcediversity.org'
    [404] link 'Merge requests'                     [18470] generic 'Public - The project
        [1102] generic '8 merge requests'                can be accessed without any
    [406] link 'To-Do List'                              authentication.'
        [1118] generic 'Todos count'  Type [1507][Byte Blaze]  [2955] image
            [1489] StaticText '5'                   [12092] generic 'blossom'
    [407] link 'Help'                                   [5288] StaticText '🌻'
    [409] link 'Byte Blaze'                         [9996] StaticText ' Code of '
        [1152] img 'Byte Blaze'                     [15143] link
    [7] main                                             'https://opensourcediversity.org'
        [302] StaticText 'Projects'                 [5065] link '28'
        [305] link 'New project'                        [845] image
        [411] link 'Yours 14'                       [5529] link '0'
        [412] link 'Starred 3'                  [4353] heading 'Checkstyle /
        [413] link 'Explore'                         checkstyle'
        [414] link 'Topics'                             [18893] link 'Checkstyle /
        [312] searchbox 'Filter by name'                     checkstyle'
        [374] button 'Name'                         [901] generic 'Public'
        [458] link 'All'                                [17524] image
```

Figure 8: A case of failed simulation where UI-SIMULATOR-R overly depends on the reference state to generate the new page.

---

You are a Web task creation AI. Assume that you have an a11y tree of this website, generate some common tasks user perform on this web.
To be successful, it is very important to follow the following rules:
1. Suppose that you've already logged in the website, and you don't want to sign out.
2. Note that we don't want the task control to focusing too much on the elements that the state contains. You should consider the functionalities behind the elements, and think of functionalities that people use.
3. The number of the tasks should be no more than {Number of Task Controls} , so just keep the most common ones. Besides, ideally the tasks are atomic independent, i.e. we don't want a certain task to be a subtask of another task, and the task shouldn't contain two subtasks like "do A and then do B".

**Example:**
**Initial state:**
{Initial State}

**Tasks:**
1. Search for a place.
2. Find directions between two places.

---

Table 10: System prompt for **First-Step Task Control Proposal**. {Initial State} is **the Initial State** for in-context example. {Number of Task Controls} limits the number of candidate task controls

You are asked to modify a web task. You will be given a description of website and original task that correspond to the website.

To be successful, it is very important to follow the following rules:

##Find the name of website and figure out what kind of website is given.
##Generate unique, diverse entity names and add to original task to make the task more specific. For example, a specific entity name for shopping website is a product name, a specific entity name for map website is a specific place name.
##Generate 15 examples. You should try to think of what kinds of terms are commonly searched.
##The search term should NOT be too complex, JUST the name. For example, we don't want "the Eiffel Tower in Paris", but just "Effiel Tower".

**Examples:**

**Website Description:** {Website Description}

**Original task:** Search for a certain product.

**##Thought:** I'll try to search for a certain product. The candidiates should be diverse. People often search foods, clothes, fruits, electric devices, toys, detergents, trip utilities on the shopping website.

**##Tasks:**
- **Foods**
    1. Search for OREO milk cookies.
    2. Search for Crisco Oil 48 Oz.
    3. Search for L'Oréal Paris Revitalift Anti-Aging Cream.
    4. ...
- **Fruits**
    1. Search for Driscoll's Strawberries.
    2. ...
- **...**

{Example #2}

{Example #3}

Table 11: System prompt for **Diverse Entity Specification**. {Website Description} is a short description of the current website for in-context example.

You are a Web task creation AI. Assume you are browsing on a website with some or no guidance. Based on the task control, the current webpage, and the browsing history, your task is to analyze the current webpage and browsing history, and continue browsing according to the task control and previous steps by giving the action on the current webpage.

Here are some requirements for output:

- You need to incorporate the following details in the **Thought**:
    - the task control (if no task control is given, just skip it),
    - what you learned from previous steps,
    - what the current webpage is like (it is best to use some elements within the webpage as evidence),
    - and which action to take next (either guided or not).
- The **Action** should strictly follow the action format provided in the action space.
- You also need to generate a **single sentence abstract** to summarize what this action does.

Note that Thought, Action and Task should not exceed one line. The summary should NOT mention the content of task control (e.g. 'Create a new project or issue' shouldn't appear in the Task in the following example), just focus on what the action does.

**Available actions:** {Input Action List}

**Example Input:**

**Task Control:** {Input Task Control}
**Current state:** {Input Current State}
**Previous steps:** {Input Previous Steps}

**Example Output:**

**Thought:** Let's think step by step. The task control is 'Create a new project or issue.'. From the previous steps, I clicked the 'New Project' button to step into the project creation page. The current webpage contains elements like "Enter your project name here" textbox with id 10 and "visibility_level" selection with id 12, which means currently I'm at the project creation page, and it has many information entries to fill in. To continue creating the project, I shall fill in the required information. I can first type a name for the new project, and the corresponding input box id is 10. 'Web platform' is a good name. I can set the third parameter to 1 to press enter to submit.
**Action:** `type [10] [Web platform] [1]`
**Task:** Type "Web platform" as the name for new project, and press enter to submit it.

Table 12: System prompt for **Thought & Action Generation**. {Input Action List} is available action space. {Input Task Control}, {Input Current State}, and {Input Previous Steps} are **Current Step Task Control, State, and Step History** for in-context example.

You are asked to generate web task controls for the next step that are unrelated to visual adjustments. You will be provided:

- Elements that are commonly used in current website.

- Original task control

- Previous steps you have taken.

You need to follow the following rules:

1. AVOID task controls related to visual manipulation.

2. You have completed the Original task control in previous steps. You should make use of the given elements to propose reasonable new task controls to extend current trajectory.

3. Pay attention to "Newly appeared elements", elements that appear in the new state compared to the last step. You should propose as many task controls based on the interactions with newly appearing elements as possible.

4. The new task controls SHOULD be consistent with previous steps and can form a single complete task rather than multiple independent tasks. E.g., we don't want to search for one place and then explore related or nearby places.

5. The task controls should be diverse. We don't want task controls doing the same thing but on different entities. For example, we don't want to have both view the detail of A and then view the detail of B in the response.

6. Only give no more than  task controls, just make sure you are proposing the most common ones.

7. You should strictly follow the output format in the example.

8. Note that you cannot interact with a StaticText or generic. Interacting with it wouldn't cause any effect. So it is best not propose task controls related to them. Also we want the tasks can be done on the computers.

**Example:**
**Original task control:**
Search for 'PHILIPS H6509 Wireless Headphones, Over-Ear Bluetooth Headphones with Noise Canceling Pro'
**Previous steps:**
["Search 'PHILIPS H6509 Wireless Headphones, Over-Ear Bluetooth Headphones with Noise Canceling Pro' "]

**##Thoughts:**

Let's think step by step. The original task control is to search 'PHILIPS H6509 Wireless Headphones, Over-Ear Bluetooth Headphones with Noise Canceling Pro', and I have completed the original task control. The current webpage shows the search results related to the 6S Wireless Noise Canceling Hi-Fi Headphones.
I need to take the next step that is consistent with the first step searching. The newly appeared elements include the detail links of the product I want, together with functionalities like "Add to Cart", "Add to Wish List", "Add Your Review", "Add to Compare", etc.
I can check the most relevant search result for its details. I can also choose to interact with the product via functionalities like adding it to the cart, wishlist, etc.

**##Task Controls:**

1. View more details about 'PHILIPS H6509 Wireless Headphones, Over-Ear Bluetooth Headphones with Noise Canceling Pro'

2. Add 'PHILIPS H6509 Wireless Headphones, Over-Ear Bluetooth Headphones with Noise Canceling Pro' to cart

3. ...

Table 13: System prompt for **Step-Wise Task Control Proposal** in later steps.

You are a Web task creation AI. Assume you have step history, and try to summarize a high-level intent.
The step history is in the format of "action: step summary".
You should pay special attention to the content within action (e.g. the content that is typed in), and the summarized task should reflect such content.

Here are some requirements for summarization:

- The high-level intent should faithfully follow the action sequence.

- The high-level intent should be succinct and consistent with each single step.

- Ignore unnecessary contexts and intermediate steps. The Task should only include the high-level goal of the trajectory.

**Example**:

**Input:**
**Previous steps:**

> `click[7809]:` Access the help section for GitLab.
>
> `click[482]:` View the 'Contact Support' page to get help with GitLab issues.
>
> `type[361][Bug:  cannot edit account detail][1]:` Enter a subject for your support request.
>
> `type[5882][Account Issue][1]:` Enter "Account Issue" in the subject field for the support request.
>
> `click[3605]:` Select the type of support needed as "Technical Support".

**Thought:**
> From steps history, the first two steps open 'Contact Support' for GitLab. The following two steps enter "account issue" for further detail. Last step select "Technical Support" as support type.

**Output:**
> **Task:** Submit a Technical support request to GitLab regarding account issue.

{Example #2}

Table 14: System prompt for **Task Summarization** in trajectory wrapping process.

You are a web behavior analyst. Assume you have collected a series of thoughts on actions taken during navigation on the web. The thoughts are not really purposeful, or not focusing on the given goal.

Your task is to rewrite each thought to make them fit the given task goal.

You should follow the rules when rephrasing thoughts:

1. The rewritten thought should consider what the current page is about and mention the goal, analyze what should be done now to complete the goal, and explain why this action is appropriate in the current step.

2. You should strictly keep the entity, the analysis on the webpage and previous steps, and the action in the original thought unchanged — just adapt the wording to make it reasonable for the goal. Ignore task control-related sentences in the original thoughts, and focus only on how to complete the given goal.

3. Please guarantee that each action appears in its corresponding rewritten thought.

**Example:**

**Original thoughts:**

- Thought 1: According to the task control, I need to view account information. The current webpage displays my account details, including "Contact Information", "Default Billing Address", and "Default Shipping Address". The "Account Information" text is prominently featured, and there is an "Edit" button linked to my account information. To proceed, I will click on the "Account Information" link to view the complete details of my account. In summary, the next action I will perform is click [3706].

- Thought 2: Moving forward as per the task control "Access the 'Payment Methods" section to add a new credit card for future purchases." From the previous step, I accessed "Account Information". Currently, there are several buttons in the account management section, including one labeled "Payment Methods" (id 1523), which I need to click to reach the section for adding a new credit card. In summary, the next action I will perform is click [1523].

- Thought 3: ...

**Goal:** Add a new credit card with the number "4111111111111111" and expiration date "12/25" to the account information.
**Actions:** `click [3706], click [1523], ...`

**Rewritten thoughts:**

- Thought 1: Let's think step-by-step. The current webpage displays my account details, including "Contact Information", "Default Billing Address", and "Default Shipping Address". The "Account Information" text is prominently featured, and there is an "Edit" button linked to my account information. This is the first step. In order to add a new credit card with the number "4111111111111111" and expiration date "12/25" to the account information, I will click on the "Account Information" link to view the complete details of my account. In summary, the next action I will perform is click [3706].

- Thought 2: Let's think step-by-step. From the previous step, I accessed "Account Information". Currently, there are several buttons in the account management section, including one labeled "Payment Methods" (id 1523), which I need to click to reach the section for adding a new credit card. In summary, the next action I will perform is click [1523].

- Thought 3: ...

Table 15: System prompt for **Thought Rewriting** in in trajectory wrapping process.

You are a teacher who is writing quiz to test your students on the reading and understanding of webpage.
The webpage should be either:

- demonstrating detailed information for one object, or

- containing information for multiple objects, and some of them are different and comparable. First, analyze the webpage.

If you think the webpage is demonstrating detailed information for one object, put "Yes" in "Answer". Otherwise put "No".
Second, you need to find relevant information that is useful for making quiz from the current webpage, such as specific numbers, ranking, entity names. If you think the webpage is wrapping information for multiple objects in a table, the questions you ask should be concentrating on the comparison of the information, or features that are in common.
Note: You need to specify the full name of entity in each question. Don't use terms like "this page", "this item". Don't ask questions about layout, e.g., buttons, textbox, or details that most people don't care, e.g. the contributor, the url of the site, etc. Don't always use interrogative sentences like "what" or "which." Instead, try using declarative sentences like "tell me ..." or "show me ...."

Examples:
**Webpage:**
[1] RootWebArea 'Carnegie Mellon University | OpenStreetMap' focused: True
    [14] heading 'OpenStreetMap logo OpenStreetMap'
    ...(omit for brevity)
    [627] StaticText 'University '
    [630] link 'Carnegie Mellon University, Pittsburgh, Allegheny County, 15213, United States'
    [624] link 'More results'
    ...(omit for brevity)

**Thought:** Let's think step by step. The current webpage is a search result of 'Carnegie Mellon University' on OpenStreetMap website. The searched result only contain detailed information about CMU, which means it is not applicable to compare with other thing. Thus the Answer should be "Yes", and I shall ask questions that are about details of CMU. There are details like the address of CMU, the zip code. I'd like to ask questions about the details.
**Answer:** Yes
**Questions:**

- Show me the address of Carnegie Mellon University.

- What is the zip code of CMU?

{Example #2}

Table 16: System prompt for **Reasoning Task Generation** in in trajectory wrapping process.

Given the current UI representation, the current action, the web browsing history, and the potentially relevant element:

First, think about what current window is like, and try to interpret the action.
Second, describe what the new window will be like after executing the action in a sentence. It is best to specify the roles of terms used in the description.
Third, extract key information that should be kept in mind, like price of bought product, user profile, etc. Feel free to put "None" here if you think the action won't cause any long-range influence.
Lastly, answer whether such action would lead to a totally new webpage.

Note: You've always logged in to the website. You don't need to consider the logging process when generating the new window. The intent should be detailed enough to describe what a whole webpage looks like.

**Example:**
**Thought:** Let's think step by step. Currently I'm at search results page of Google. The Google search results are general. The 'click' action is targeted on a hyperlink to 'Alan's podcast channel'. Since we have set up the age verification and pass the age limit, this action redirects the user to its homepage.
**New window:** The new window is Alan's live podcast home page.
**Key Info:** None
**Answer:** Yes
{Example #2}
{Example #3}
{Example #4}

Table 17: System prompt for **Next State Overview Prediction**.

Imagine you are a website designer. Given some previous information, the interpretation of action on last step, and description of a new website, first extract what the new website (and the domain) is, then answer the question: What sections should the webpage have, and list all of them and their functionalities, and compose elements that appear in each section in detail.

You should ONLY generate informative elements that are relevant with the description. Informative elements refer only to the main elements that contain specific, instantiated information of the webpage. For example, a paragraph with texts introducing the term "California", the link to Youtube, etc.

Pay attention to the input, which contain information that must be included in current page. Also remember you are creating content within a certain domain. Elements like "Copyright: Alhambra Palace" will never appear in a Google map webpage.

For a bunch of similar, important elements to generate (e.g. search results on a search results page), the number of such element should be at least 6.

You should generate content based on the domain, and information that reasonably appear in the domain. E.g., we should have prices information in a shopping website. Sometimes you can have section like "Other related terms", but that should never be the main content.

Note that if you think an element can be interacted with, specify that it should be a link element. E.g., you don't have to add another element "View details" or "Website" for a search item, because when clicking on the search term, we might jump to the detail of the item. Just merge their functionalities and tag the element as a link.

**Example:**

**Previous Info:** {Previous Key Info}

**Description:**
The new window displays the discussion thread page on Reddit. The title of the post is "What do you think of the new European Cup champion". User could read and interact with the thread.

**Thought:**

**Thread interaction section**

A "Comment" button for users to engage directly with the post
A "Share" button for users to share the discussion thread
An "Upvote" button and a "Downvote" button for users to express their opinions on the post

**Title section**

Title of the post: What do you think of the new European Cup champion Spain?

**Body section**

Post content: "Spain has triumphed in the ..."
Link of Post number: #10003902
Upvote: 569
...
**A comment section where users can share their thoughts, including:**
  **Comment:** "I think Spain played incredibly well! Their teamwork was on another level!"
    Link of commenter: Alice; upvote: 5; downvote: 0; 12h ago; upvote/downvote button
  **Comment:** ...

Table 18: System prompt for **Generating Rich Draft in Natural Language**. {Previous Key Info} contains some key information recorded in previous steps to boost the coherence of the content.

Given the following reference action sequences and the current action sequence, your task is to find the ones from the reference sequences that are doing almost the same thing as current action sequence. If there's no sequence in reference that does the same thing as current action sequence, then you should pay more attention to the ending steps, and choose ones that are doing the same thing in the latest steps. If the functionality of the current action history is not quite clear, just finding ones that exactly match the latest steps.
Note: You need to consider the meaning behind actions like "click", "type". E.g., click button 'Search' means doing the search on the typed term. We DONT want the output sequences to have the future steps of the current action history. You should find sequences that just stopping at the same point as the current actions. You should strictly follow the output format.

**Example:**

1. type textbox 'Search' 2. click button 'Search' 3. click link 'Dell G7 Laptop' 4. click 'Add to cart'

1. type textbox 'Search' 2. ...

**Current action sequence:**

1. type textbox 'Search' 2. click button 'Search' 3. click link 'OREO milk cookies' 4. click 'Add to cart'

**##Thoughts:**

Let's think step by step. The current action sequence does searching at first, then clicks 'OREO milk cookies' to view its details, and adds it to the cart. I should output action sequences that are doing the same thing at the ending steps.

**##Output:**

1. type textbox 'Search' 2. click button 'Search' 3. click link 'Dell G7 Laptop' 4. click 'Add to cart'

1. type textbox 'Search' press enter 2. ...

{Example #2}

Table 19: System prompt for model-based **Semantic Retriever** based on action history.

Given the following description of a GUI and the refernce GUI, create the content of a new state by taking elements and rewriting some important contents within the reference state.

Note that the description of content might be short, but you should provide corresponding essential information in the reference GUI. So you should pay attention to each element in the reference GUI.

If you think the reference GUI doesn't match the descrition, then you should generate a lot of new contents that match the description, but within the same structure.

If you think the reference GUI matches the descrition, you could copy most of the content from the reference state, and only modify a few important contents if needed. Do not try to add additional details or information in this case.

- You are only allowed to modify information related to some named entities. You are not allowed to add/remove the functionality elements in the reference state if you think it matches the description. If no named entities are in the reference state, you can make no change.

**Example input:**
**Reference state:** {Reference State}
**Description of new state:**
    The new website is a product page on Onestopshop ...

Example output:
{New Content}

Table 20: System prompt for **Retrieval-Augmented Draft Generation** . {Reference State} refers to the state retrieved for generation. {New Content} contains a list of realistic elements inherited from the reference state and newly composed content in the current context

You are a Web task creation AI. Given the a11y tree of a website, generate a common task user perform on this web, and new browsing history adapted from the given browsing history. The reference task and browsing history is based on another webpage that has the similar functionalities but with different object. So you should propose task based on content from currently given entities and content.

To be successful, it is very important to follow the following rules:
1. Suppose that you've already logged in the website, and you don't want to sign out.
2. The new task should strictly have the same task type as the reference. Don't change the task type
3. Don't change the procedure in the browsing history. Just change the entity names of objects (products, items), not functionalities.
If the reference browsing history is "None", then put "None" in the new browsing history. Don't imagine steps that are doing different things from reference browsing history.

**Example:**

**Current webpage:** {Input State}
**Reference task:** Add 'Dell G7 Gaming Laptop - 256GB' to the cart.
**Reference browsing history:**
1. Search for "Dell G7 Gaming Laptop"
2. Click 'Dell G7 Gaming Laptop - 256GB' to view its details.

**Task:** Add 'Milkman Bonus Bundle - 10 Packets Low-Fat Milk + 2 Packets Chocolate Milk with 18g Protein' to the cart.
**New browsing history:**
1. Search for "Milkman Bonus Bundle"
2. Click 'Milkman Bonus Bundle - 10 Packets Low-Fat Milk + 2 Packets Chocolate Milk with 18g Protein' to view its details.

Table 21: System prompt for **Targeted Task Variant Synthesis**. {Input State} contains details of a product ('Milkman Bonus Bundle' in this case)

Assume you are a Android mobile phone. Based on the task control, the current UI page, and the action history, your task is to analyze the current page and history, and continue browsing according to the task control and previous steps by giving the action on the current page.

Here are some requirements for output:

- You need to incorporate the following details in the **Thought**:
    - the task control,
    - what you learned from previous steps,
    - what the current page is like (it is best to use some elements within the page as evidence),
    - and which action to take next.
- The **Action** should strictly follow the action format provided in the action space.
- You also need to generate a **single sentence abstract** to summarize what this action does.

Note that Thought, Action and Task should not exceed one line. The summary should NOT mention the content of task control (e.g. 'Create a new project or issue' shouldn't appear in the Task in the following example), just focus on what the action does.

**Available actions:** {Input Action List}

**Example Input:**

   **Task Control:** {Input Task Control}
   **Current state:**

   Element     0:        UIElement(text=None,      content_description=Create     contact,
class_name=android.view.View,      bbox=None,      bbox_pixels=BoundingBox(x_min=0,
x_max=1080,      y_min=0,      y_max=2400),      hint_text=None,      is_checked=False,
is_checkable=False,      is_clickable=False,      is_editable=False,      is_enabled=True,
is_focused=False,    is_focusable=False,    is_long_clickable=False,    is_scrollable=False,
is_selected=False,     is_visible=True,     package_name=com.google.android.contacts,     re-
source_name=com.google.android.contacts:id/background_container,      tooltip=None,      re-
source_id=None, metadata=None))
   Element 1: UIElement(text=None, content_description=None, class_name=...)
   Element 2: UIElement(text=First Name, content_description=James, class_name=...)
   Element 3: UIElement(text=Last Name, content_description=Brown, class_name=...)
   Element 4: UIElement(text=Phone Number, content_description=None, class_name=...)
   Element 5: UIElement(text=Email Address, content_description=None, class_name=...)
   Element 6: UIElement(text=Save, content_description=None, class_name=...)
   Element 7: UIElement(text=Cancel, content_description=None, class_name=...)
   ...

   **Previous steps:** {Input Previous Steps}

**Example Output:**

   **Thought:** Let's think step by step. The guide is 'Create a new contact for "James Brown". From previous steps, I opened the 'Contacts' app, started the creation process and typed the first and last name. The current page shows that I've successfully typed the First and last name, and I also need to fill in details like phone number, email address. Since the guide doesn't provide the phone number, I should give a realistic phone number here, like "718-099-5256". To continue creating the contact, I shall type "718-099-5256" to the Phone number.
   **Action:** `input_text [4][718-099-5256]`
   **Task:** Type "718-099-5256" as the phone number.

Table 22: System prompt for **Thought & Action Generation** for **AndroidWorld** trajectory collection. {Input Action List} is the action space. {Input Task Control}, {Input Current State}, and {Input Previous Steps} are **Current Step Task Control, State, and Step History** for in-context example.

