# OpenReview forum: "LLMs as Scalable, General-Purpose Simulators For Evolving Digital Agent Training"
_ICLR.cc/2026/Conference — Submitted to ICLR 2026_

### Official Review · Reviewer_Fc9J · 2025-10-20

**Soundness:** 2
**Presentation:** 2
**Contribution:** 2
**Rating:** 2
**Confidence:** 5

**Summary:**

This paper introduces UI-SIMULATOR, a scalable trajectory synthesis paradigm that uses LLM-based digital world simulators to synthesize diverse UI trajectories. The authors further propose UI-SIMULATOR-GROW, which enables the continuous improvement of synthesized data. Experimental results on WebArena and AndroidWorld demonstrate the promise and robustness of this trajectory synthesis method.

**Strengths:**

1. The proposed scalable UI trajectory synthesis paradigm, UI-SIMULATOR, offers an interesting new direction for synthesizing data for GUI agents.
2. The UI-SIMULATOR-GROW extension effectively improves the model's utilization of synthetic data, ensuring stable performance gains.

**Weaknesses:**

1. Figure 2 is rendered poorly and has several clear issues. First, the screenshot in the middle is too small, even when I zoom the image to fill the entire screen, the text within it remains difficult to read. Second, the figure fails to clearly distinguish between the Retrieval-Free and Retrieval-Augmented Simulator methods. The latter should logically collect higher-quality trajectory data, yet the figure depicts identical actions and states for both simulators, failing to illustrate the benefit of using prior experience. Third, the figure lacks sufficient informational content. It is a missed opportunity to present the complete architecture of UI-SIMULATOR, leaving the paper without a central, overarching diagram to visually anchor the methodology described in the text.
2. Sections 3 and 4 could be merged. Section 3 provides a conceptual and design overview of the entire process of building a digital world simulator within UI-SIMULATOR, while Section 4 describes the specific data collection methodology. Both sections are fundamentally about explaining UI-SIMULATOR. Presenting them separately creates a distinct sense of fragmentation. Merging them would allow the two perspectives to complement each other, giving readers a more profound and cohesive understanding of the system. Furthermore, Section 3 does not seem to warrant its current length, as it does not involve extensive methodological design.
3. Some experimental results are puzzling. First, the reported result for Qwen-2.5-7B-Instruct on AndroidWorld is worse than Qwen-2-VL-7B, and is reported as zero. The official Qwen-2.5-7B-Instruct result on AW is 25.5, which I have also personally verified. It is unclear if this discrepancy stems from an issue with the authors' action space or coordinate conversion. Second, the scores reported for baselines like NNetNav, OS-Genesis, and GUIMid appear incorrect. On AndroidWorld, OS-Genesis scores around 17 and GUIMid around 21. On WebArena, NNetNav has a SR of 16.3, and the other two score around 10. I suspect the authors may have reproduced these results under their own setting; if so, the comparison would be unfair. Additionally, including GUIMid as a baseline is inappropriate, as it is not a data synthesis method.
4. Although the proposed method is presented as SOTA in the provided tables, its performance still lags significantly behind the latest work on GUI agents. For instance, the top scores on AndroidWorld have exceeded 70[1], yet UI-SIMULATOR's result is nearly 50 points lower than a model of a similar scale like GUI-Owl-7B (66.4)[1]. While I understand the limitations of synthetic data, such a large performance gap makes the proposed method less competitive.

[1] Ye J, Zhang X, Xu H, et al. Mobile-agent-v3: Fundamental agents for gui automation[J]. arXiv preprint arXiv:2508.15144, 2025.

**Questions:**

1. Line 214 mentions a "first stage" of the data collection process, but I could not find any mention of a "second stage" or "next stage" in the text of Section 4.1. Does it refer to the "step-wise guided rollout process and a final trajectory wrapper" mentioned in the last line of the section? The phrasing here is not very clear.
2. The "Step-wise guided rollout process" seems very similar to the Explorer[1] method: both propose a high-level task from an initial screen, have the agent interact with the environment to iteratively update goals, use a verifier, and summarize the task at the end. Could you please provide a detailed explanation of the differences between your method and Explorer, as well as the advantages of your approach?
3. LLM-based trajectory generation relies heavily on A11y Tree, which are often noisy. Furthermore, the A11y Trees for some websites or applications can be incomplete. I would like to ask how the authors address these two common problems.

[1] Pahuja V, Lu Y, Rosset C, et al. Explorer: Scaling exploration-driven web trajectory synthesis for multimodal web agents[J]. arXiv preprint arXiv:2502.11357, 2025.

---

### Official Review · Reviewer_t5K3 · 2025-10-31

**Soundness:** 3
**Presentation:** 3
**Contribution:** 2
**Rating:** 4
**Confidence:** 5

**Summary:**

This paper introduces **UI-SIMULATOR**, a framework that treats large language models (LLMs) as **digital world simulators** for generating large-scale UI interaction trajectories to train general-purpose digital agents. The framework includes a world simulator for state transitions, a guided rollout process for multi-step task exploration, and a trajectory wrapper for refining reasoning and task instructions.

A key extension, **Retrieval-Augmented Simulation (UI-SIMULATOR-R)**, allows the model to reference a few real-environment samples during generation, making simulated UI states more realistic and domain-consistent. The authors further propose **UI-SIMULATOR-GROW**, a targeted scaling strategy that selects mid-difficulty tasks and synthesizes diverse variants for efficient model improvement.

Experiments on **WebArena** and **AndroidWorld** show that simulated data—especially with retrieval augmentation—matches or surpasses real-environment training, demonstrating the potential of LLM-based simulation as a scalable, data-efficient paradigm for digital agent training.

**Strengths:**

The paper proposes a **well-structured and carefully engineered pipeline** for scalable digital agent training based on LLM-driven UI simulation. The design integrates multiple complementary components—guided rollouts, retrieval-augmented simulation, trajectory refinement, and targeted scaling—each with clear motivation and technical soundness.

The **implementation quality and modular design** are impressive, and the **ablation studies are thorough and informative**, effectively illustrating the contribution of each component. While the overall performance gains are moderate, the system demonstrates solid empirical grounding and provides a valuable, reproducible framework for future research on LLM-based digital world simulation.

**Weaknesses:**

1. The pipeline relies on **a relatively weak base model (GPT-4o-mini)** as the core simulator. Many auxiliary modules (guided rollouts, trajectory rewriting, targeted scaling, etc.) appear necessary to compensate for its limited capability. However, if stronger LLMs (e.g., GPT-5-High, Claude-4.5-Sonnet) were used, much of this complexity might become unnecessary. The overall contribution would benefit from a clearer discussion of how the framework scales with model strength and whether the modular design remains essential under more capable simulators.

2. The **performance improvements are rather limited**, especially considering that current baselines on WebArena (≈50) and AndroidWorld (≈60+) are already relatively high. Models trained purely on the synthesized data achieve only around 10-point success rates, which questions the practical impact and realism of the generated data. The paper would be stronger with a deeper analysis of why synthetic trajectories fail to translate into higher downstream gains, or with qualitative evaluations showing where the simulated data still diverges from real interaction patterns.

**Questions:**

1. The authors may consider **re-evaluating the pipeline with stronger base models** (e.g., GPT-5-High, Claude-4.5-Sonnet, or Llama-3-70B) to examine whether the proposed augmentation modules—such as guided rollouts and trajectory rewriting—remain necessary or still contribute meaningfully when the simulator itself is more capable. This would help clarify whether the framework’s complexity is inherently valuable or primarily compensatory for weaker models.

2. It would also be interesting to **test the generated data within more advanced agent frameworks**, such as **AgentLab**, or with stronger open-source agents (e.g., **Qwen2.5-32B**, which can reach ≈20 success rate on WebArena). Demonstrating improvements under these more competitive setups could better validate the real-world utility and transferability of the synthesized data.

---

### Official Review · Reviewer_kxKf · 2025-10-31

**Soundness:** 2
**Presentation:** 2
**Contribution:** 2
**Rating:** 4
**Confidence:** 3

**Summary:**

The paper propose UI-SIMULATOR, a method for producing synthetic models of UI states and transitions. Additionally, the paper proposes UI-SIMULATOR-GROW, a method that uses UI-SIMULATOR combined with an instruction generator to collect synthetic trajectories to improve UI agents. The paper shows that these techniques improve agent performance on WebArena and AndroidWorld.

**Strengths:**

* Using LLMs to generate synthetic environments for digital agents is an interesting area of exploration. This paper proposes an interesting perspective on using LLMs to generate world models of this form. Rather than generating the source code to simulate web or mobile apps, the LLM generates states and transitions. I think analyzing the effectiveness of this approach is quite interesting for the community.
* The empirical results show improvements on WebArena and AndroidWorld relative to baselines.

**Weaknesses:**

The core novel method that this paper explores is using LLMs to generate synthetic UI states and transitions -- a nice topic! The extension to use a given environment to synthesize instructions and trajectories has been explored by prior work, and the contributions there were less clear. I think the paper could be improved by focusing the presentation and analysis on the core contribution related to generating synthetic environments.

* Presentation: Section 3 seems to describe the core contribution, but defers most of the detail to the appendix. The details of Figure 2 are not legible. I did not follow the summary of the combination of rule-based and LLM-generated trajectories without following the reference to Appendix C, and similarly did not follow how retrieval-augmentation is used (deferred to Appendix D). As this is the core contribution, I think it would be useful to present this aspect of the method more clearly. The other aspects of the overall UI-SIMULATOR-GROW recipe could have potentially adopted methods from prior work, e.g. https://arxiv.org/abs/2410.02907. In short, I thought section 3 should be expanded and sections 4 and 5 could be reduced if relying more on methods from prior work, to better focus the contribution.
* Analysis: It was difficult to evaluate the core contribution (generating synthetic environments) given the complexity of the overall recipe. There is some discussion in section 7.1, but it seems the quantitive results are deferred to the appendix. How do synthetic environments of this form compare to simply asking the LLM to generate frontend environments with transitions described in the source code? I liked the brief mention of a comparison with exploring the test environments directly and the anecdote that synthetic databases can lack coverage over all search queries, but wanted more of this quantitative and qualitative analysis in the main paper.
* Benchmarks: The overall results are so low (success rates <15%) on WebArena and AndroidWorld (much below SOTA) that I wonder if benchmarks with more dynamic range would be more useful. (It's fine to not be SOTA but the small effect sizes make the results harder to interpret)

Summary: I thought the presentation and analysis in the main paper did not sufficiently focus on the core contribution of developing synthetic environments following the proposed method. I think it would be a more useful paper to focus on 1-2 key research questions rather than introduce a complex pipeline with many novel components that are not individually well justified.

**Questions:**

See weaknesses above.

---

### Official Review · Reviewer_mEH9 · 2025-10-31

**Soundness:** 2
**Presentation:** 3
**Contribution:** 3
**Rating:** 2
**Confidence:** 4

**Summary:**

This paper proposes UI-SIMULATOR, a scalable method that uses LLMs to simulate digital UI environments and generate diverse, high-quality training trajectories for agents—eliminating the need for costly real-world data collection. It also introduces UI-SIMULATOR-GROW, a targeted data scaling strategy that prioritizes the most useful tasks to improve data efficiency. Experiments on WebArena and AndroidWorld show that agents trained with this approach match or outperform those trained on real environments, even when using smaller models, demonstrating that LLMs can act as effective, scalable simulators for digital agent training.

**Strengths:**

1. **Novel Application of LLMs for Digital UI Simulation**
   The work creatively positions LLMs as world models capable of generating realistic and structured UI transitions, removing the need for traditional real-environment interaction.

2. **High Data Efficiency with Competitive Performance**
   The proposed methods deliver strong task success rates using significantly fewer training samples and smaller model sizes, showcasing impressive efficiency in both data and compute.

3. **Robust and Domain-Agnostic Generalization**
   The approach demonstrates strong robustness to perturbed environments and generalizes effectively across both web and mobile UI tasks, indicating broad applicability.

**Weaknesses:**

1. The paper writing is not sufficiently clear on some figures.

2. Experiments need to be further clarified.

3. Method claiming on diversity remains quentionable.

**Questions:**

1. The method's main contribution is to treat LLM as similarotrs to collect data.
How good are current LLMs treated as world models? Do authors treat this point for granted or have some prior investigation?

2. Figure 2 is too vague to read. I would suggest to highlight only the core part of the code (Instead of using tiny font to fit all codes into the fiture)

3. From my side, it is not appealing to say your trained 8B model is better than 70B model because your model is specially in-domain trained. Authos are proposing a data collection pipeline, will this pipeline fits different kinds of base models? The justification of this pipeline on more base models as well as teacher models are much more important yet lacking in current paper.

4. What is the performance of your teacher model (gpt-4o-mini) ? Will you forsee your trained model surpass the teacher model?

5. Can your collected data further boost the performance of your teacher model ? You can SFT your OpenAI teacher with your constructed data (FYI: https://platform.openai.com/docs/guides/supervised-fine-tuning).

---

### Meta-Review · Area_Chair_Dw1K · 2026-01-01

**Summary:**

All reviewers agree that the submission has major issues with its technical presentation around its method and experiment analysis, while the overall contribution remains unclear. In addition, R-kxKf and R-t5K3 also raised that the performance improvements on WebArena and AndroidWorld are a bit limited and the absolute scores are much lower than SoTA approaches on these benchmarks. Furthermore, the relatively low success rates (<15%) also make it hard to interpret the overall results. Given these issues and the fact that the authors did not provide a rebuttal, the recommendation is a reject.

**Reviewer Concerns:**

All concerns remain unaddressed.

**Reviewer Scores:**

N/A (the authors did not provide a rebuttal).

---

### Decision · Program_Chairs · 2026-01-26

Reject